# Interlayer and Intralayer Excitons in *AlN*/*WS*_2_ Heterostructure

**DOI:** 10.3390/ma15238318

**Published:** 2022-11-23

**Authors:** Claudio Attaccalite, Maria Stella Prete, Maurizia Palummo, Olivia Pulci

**Affiliations:** 1Centre Interdisciplinaire de Nanoscience de Marseille UMR 7325 Campus de Luminy, CNRS/Aix-Marseille Université, CEDEX 9, 13288 Marseille, France; 2European Theoretical Spectroscopy Facilities (ETSF); 3Dipartimento di Fisica, Universitá di Roma Tor Vergata, and INFN, Via della Ricerca Scientifica 1, I-00133 Rome, Italy

**Keywords:** exciton, 2D materials, optical properties, ab-initio, DFT, GW, BSE

## Abstract

The study of intra and interlayer excitons in 2D semiconducting vdW heterostructures is a very hot topic not only from a fundamental but also an applicative point of view. Due to their strong light–matter interaction, Transition Metal Dichalcogenides (TMD) and group-III nitrides are particularly attractive in the field of opto-electronic applications such as photo-catalytic and photo-voltaic ultra-thin and flexible devices. Using first-principles ground and excited-state simulations, we investigate here the electronic and excitonic properties of a representative nitride/TMD heterobilayer, the $AlN/WS_{2}$. We demonstrate that the band alignment is of type I, and low energy intralayer excitons are similar to those of a pristine $WS_{2}$ monolayer. Further, we disentangle the role of strain and *AlN* dielectric screening on the electronic and optical gaps. These results, although they do not favor the possible use of *AlN*/$WS_{2}$ in photo-catalysis, as envisaged in the previous literature, can boost the recently started experimental studies of 2D hexagonal aluminum nitride as a good low screening substrate for TMD-based electronic and opto-electronic devices. Importantly, our work shows how the inclusion of both spin-orbit and many-body interactions is compulsory for the correct prediction of the electronic and optical properties of TMD/nitride heterobilayers.

## 1. Introduction

Guided by graphene rise [1], a broad family of two-dimensional (2D) materials with different electronic and optical properties [2] is currently being studied for fundamental research and also for a plethora of envisaged device-oriented applications [3]. Due to their flat nature, 2D materials show unique potential to fabricate flexible and ultra-thin devices with the hope of reducing the costs of production and improving performances. Layered heterostructures offer a unique playground to engineer their electronic and optical properties [4,5] thanks to the availability of metallic, semiconducting and insulating monolayers (MLs) and to the possibility to stack them with any order and orientation using Van der Waals growth [6]. Hence, there are essentially no epitaxial lattice-match requirements typical of 3D materials.

Two emerging classes of 2D materials, Transition Metal Dichalcogenides (TMDs) and group-III nitrides, are particularly attractive for their sizable band-gaps, strong light–matter interaction and their interesting excitonic properties, which hold promise for their use in opto-electronic applications, such as ultra-thin and flexible photovoltaics (PVs) or photo-catalytic cells and light-emitting diodes (LED).

After the discovery of the indirect to direct gap behavior reducing the thickness of MoS${}_{2}$ to a monolayer form [7,8], the structural, electronic and optical properties of two-dimensional TMDs have been widely investigated at the theoretical level by means of Density Functional Theory (DFT) and refined excited state methods (namely GW and BSE) [9,10,11,12]. At the same time, an increasing number of DFT studies [13,14] has been published regarding 2D nitrides, and several predictions based on GW and BSE methods exist in the literature [15,16,17,18]. TMD vdW heterostructures (HTs) have been widely investigated in recent years, and also some studies have appeared in the literature on their combination with other 2D materials [19,20,21]. However, much less attention has been dedicated to HTs obtained by combining 2D TMDs and nitrides.

In particular, it is of interest to identify if a type I or type II band alignment is present and how the electronic and optical properties of an isolated TMD change in the presence of a nitride (beyond the normally used hBN) substrate.

Indeed, while a type I band alignment can be a good prerequisite for strongly bound excitons associated to efficient light emission, a type II alignment favors the formation of long-lived interlayer excitons and enables their ultra-fast charge transfer [22]. This is interesting not only for the investigation of novel excitonic physics, such as quantum Bose gases [14,23,24], but also for the design of innovative opto-electronic devices.

Then, motivated also by a recent experimental work [25] that shows how 2D hexagonal AlN can be grown by using ALD on TMD, resulting in a gate dielectric material alternative to h-BN [18] and by other works [26,27,28] showing the growth of hexagonal group III-nitrides on TMDs, our goal here is to investigate the structural and opto-electronic properties of a prototype bi-layer composed of $AlN/WS_{2}$. Our aim is to focus on the role of many-body effects, which are expected to be of primary importance due to the reduced dimensionality and to the low dielectric screening. It is worthwhile to mention that a previous study Liao and collaborators [29] proposed this class of heterobilayers for the next generation of ultra-thin flexible opto-electronic devices (with a focus on AlN(GaN)/MoS${}_{2}$), but this remains at the single-particle DFT level of approximation.

## 2. Methods and Computational Details

Our first-principles calculations are based on DFT and many-body perturbation theory (MBPT). For DFT structural calculations, we have used the PBEsol exchange-correlation functional [30] and DOJO pseudopotentials [31] within the *Quantum Espresso* code [32]. Van der Waals corrections [33] are applied on top of the PBEsol functional in order to take into account the weak interaction between the layers in the $AlN/WS_{2}$ heterostructure. A (24×24×1) **k**-point mesh and an energy cutoff of 120 Ry have been used. In order to avoid spurious interaction between adjacent images, a supercell of 35.5 Bohr thickness is used, and a cutoff on the Coulomb interaction is applied [34]. The electronic band structure and optical absorbance spectra are calculated taking into account spin-orbit coupling (SOC), both at the DFT level of approximation and beyond. The MBPT simulations are performed using the Yambo code [35]; namely, we first corrected the DFT band structure by doing G${}_{0}$W${}_{0}$ calculations [36] and then solved the Bethe Salpeter equation (BSE) [37] to obtain the optical properties taking into account excitonic and local-field effects as well as the full spinorial nature of the electronic wavefunctions [38]. The parameters of convergence are the following: 90 Ry cutoff for the wave functions, a (33×33×1) **k**-point mesh, both for the exchange and correlation part of the self-energy, 5 Hartree cutoff for the dielectric matrix and 500 bands. In order to speed up convergence with the number of empty bands, we use the a terminator both in G and W [39]. Note that for W, we used the Godby–Needs plasmon pole model [40], which is not based on sum-rules and therefore requires a lower number of conduction bands to converge [41]. For the calculation of the BSE optical spectrum, we have included the eight highest valence bands and the eight lowest conduction bands, and the same **k**-point sampling of the G${}_{0}$W${}_{0}$ calculations.

## 3. Results

### 3.1. Crystal and Electronic Structures of $AlN/WS_{2}$ vdW Heterostructure

The relaxed lattice parameters are 3.11 Å and 3.16 Å for isolated AlN and $WS_{2}$ monolayers, respectively. The lattice mismatch is less than 2%, which is a good value for constructing an $AlN/WS_{2}$ heterostructure based on a 1 × 1 cell periodicity. As a starting point, we chose a lattice parameter intermediate between those of the layers; then, we relaxed the combined structure, obtaining a final lattice parameter of 3.142 Å. For this HT, we consider six configurations, as illustrated in Figure 1, where different rotation angles and stacking between the adjacent sheets have been selected. The cell and the atomic positions for all configurations are then relaxed using a 24×24×1
**k**-point sampling, 120 Ry cutoff for the wave function and a cutoff on forces of $10^{-4}$ Ry/au.

In Table 1, we list the calculated energy difference between the total energy of various stacking configurations and the most stable one, the interlayer distance between AlN and $WS_{2}$ layers, as well as the W-S and Al-N bond lengths for the $AlN/WS_{2}$ heterostructures. The energy difference ΔE is defined as $\Delta E=E-E_{0}$, where E${}_{0}$ is the total energy of the most stable configuration and E is the total energy of each configuration. The calculated most stable structure (with ΔE = 0 eV) for $AlN/WS_{2}$ is the **(a)** configuration, where N(Al) atoms lie on top of W(S) (see Figure 1). In order to investigate the thermodynamic stability of this configuration, we calculate a stacking energy value of −12.45 meV, according to
(1)$$E_{stack}=E_{total}^{AlN/WS_{2}}-E_{total}^{WS_{2}}-E_{total}^{AlN}$$
where $E_{total}^{AlN/WS_{2}}$, $E_{total}^{WS_{2}}$ and $E_{total}^{AlN}$ represent the total energies of the ${AlN/WS}_{2}$ heterostructure, of $WS_{2}$, and of *AlN* monolayers, respectively. The negative value indicates that this heterostructure is energetically favorable and could be experimentally realized.

For isolated *WS*${}_{2}$ and AlN, our electronic DFT results are consistent with the existing literature [42,43,44]. It is well known that the direct/indirect gap nature of TMD monolayers is strongly dependent on the lattice parameters and small changes in the approximations used in the DFT calculations. For this reason, in this work, we used a modified GGA exchange functional [33], together with nonlocal correlation for the second version of the van der Waals density functional of Lee et al. [45]. This functional has shown very good performance in two-dimensional heterostructures [33].

*WS*${}_{2}$ has a direct DFT gap of 1.64 eV located at the **K** point with a spin-orbit splitting of 0.036 eV for the first conduction bands. The AlN monolayer shows an indirect DFT gap of 2.97 eV with CBM and VBM located at Γ and **K** points, respectively, in agreement with previous calculations [16].

When we combine the two systems, forming the $AlN/WS_{2}$ heterostructure, the final DFT electronic bands present an indirect gap of E${}_{gap}^{KQ}$ = 1.44 eV, between K and **Q**, while the direct one is located at K, E${}_{gap}^{KK}$ = 1.50 eV (see 3rd row of Table 2 and Figure 2). These gaps are smaller than those of the separated layers, and although this could be quite unexpected for a vdW heterostructure, it has been reported several times in the literature [46,47,48,49] and suggests a type II junction.

Indeed, calculating the projected density of states (PDOS), it is shown that the CBM and VBM states are localized on different monolayers of the vdW heterostructure (see Appendix A).

As seen more clearly in Figure 3, the HOMO, located at **K**, mainly consists of the N $p_{z}$ orbitals, while the LUMO, located at **Q**, is mainly due to W $d_{z}$ orbitals.

The interaction between the two layers also increases the spin-orbit splitting of the conduction bands at **K** to 0.042 eV with respect to the isolated $WS_{2}$. This is not unexpected because it has already been shown that charge transfer and strain can modify the spin-orbit splitting of two dimensional materials [50,51].

Concerning the DFT level of approximation, we can then affirm that the $AlN/WS_{2}$ is a type II heterostructure, hence suggesting it could have promising properties for charge separation in photo-voltaic/photo-catalytic devices [29].

In order to confirm this interesting result, we need to go beyond DFT to better take into account exchange and correlation effects. We hence applied the perturbative one-shot GW method to $AlN/WS_{2}$ and to the separated layers. It is worthwhile to remember that in this work, in contrast to [29], SOCs are included both in DFT and in MBPT calculations [38].

### 3.2. Quasi-Particle Effects

We report in Table 2 the direct and indirect gaps at the different levels of approximation of all the three systems *AlN*, $WS_{2}$ and $AlN/WS_{2}$. We discuss here the results for the relaxed $AlN/WS_{2}$ structure (third row in Table 2), while the constrained structure (last row in Table 2) is discussed in Section 3.3.1.

The GW correction in AlN opens the gap up to 5.61 eV without changing its position in *k* space. In $WS_{2}$, instead, the calculated GW corrections open the gap and make WS${}_{2}$ an indirect gap material with the minimum gap located between **K** = (13, 13, 0) and **Q** = (1066, 1055, 0) [43]. The GW also reduces the spin-orbit splitting of the lowest conduction bands at **K** to 0.016 eV.

Now, we move the $AlN/WS_{2}$ heterostructure. The accurate GW electronic band structure of the relaxed bilayer is displayed in Figure 2. The inclusion of the GW corrections reduces the SOC splitting at **K** to 0.028 eV. Interestingly, the quasi-particle corrections induce an interchange of the bands close to the K point. The DFT highest occupied double degenerate valence bands (HOMO), located on the nitride layer, are more down-shifted by the GW corrections than the double-degenerate valence bands (HOMO-1), located on $WS_{2}$, which then become the new VBM. In other words, the valence band which is the highest in DFT (HOMO) becomes the HOMO-1 in GW, and the band which in DFT is HOMO-1 becomes the new HOMO in GW. This band exchange is due to two reasons: first, the different nature of the top valence bands belonging to the *AlN* and $WS_{2}$, where the former are mainly formed from *p* orbitals of the N atom while the latter originate from the *d* orbitals of the W atom, and so they acquire a different GW correction, as happens often in molecular systems [52]; second, the different screening felt by the electrons in *AlN* or in $WS_{2}$—in fact, the screening is less effective in *AlN* (it has a larger gap), and this causes a larger quasi-particle correction on the *AlN* states with respect to $WS_{2}$ states. Therefore, while at the DFT level a type II heterostructure is obtained, the GW corrections modify this picture and give a type I system, where both top valence and bottom conduction bands belong to the $WS_{2}$ subsystem.

In this context, it is worthwhile to point out that the type I band offset can be obtained only considering both SOC and many-body effects [38]. Indeed, in a recent work by Yeganeh et al. [53], a type II band alignment at the G${}_{0}$W${}_{0}$ level but without the inclusion of SOC was found.

### 3.3. Optical Properties of $AlN/WS_{2}$

The optical response of the HT is investigated at two different levels of theory: within the independent-QP approach at the G${}_{0}$W${}_{0}$ level and with the inclusion of local-fields and excitonic effects by solving the BSE. In Figure 4, we present the in-plane optical absorbance for the two cases.

In order to understand the optical properties of the heterostrucure, we start with the study of the two separated layers.

The optical properties of free standing AlN, in the energy range we are interested in, are dictated by a single excitonic peak at 4.75 eV that is double degenerate; see the bottom panel of Figure 4 in a dashed blue line. This direct exciton is strongly bound and has a large oscillator strength and a very small radius (see Table 3). Because of the large GW direct gap (6.27 eV), no optical response is visible at the independent-QP level in the range here considered (0–5.5 eV, see Figure 4 upper panel).

The optical response of $WS_{2}$ (green dashed line in Figure 4) is more complex. The schematic representation of the $WS_{2}$ excitonic levels is shown in Figure 5. The lowest exciton at 2.075 eV is spin-forbidden dark and double degenerate, generally called *A*${}_{D}$. The first bright exciton *A*, also double degenerate, is at 2.125 eV. Then, the *B* exciton, due to spin-orbit split bands, is at 2.46 eV. Between the *A* and *B* excitons, a series of small peaks, called *A*${}^{\prime }$, coming from transitions near $K,K^{\prime }$ and with a very small dipole matrix elements, are present. These results are in agreement with theoretical predictions and experimental data [8,54,55,56]. At higher energy, other excitons (normally called *C* and *D*) due to transitions in the band-nesting Γ−K region are visible. According to the literature [57], they are more difficult to converge and are also more affected by the electron–phonon interaction, which is not included in the present work. The $WS_{2}$ exciton levels (see Figure 5) are in good agreement with the previous calculation of Marsili et al. [38], and the small differences are due to the different pseudo-potentials used in the calculations.

We now move to the analysis of optical properties of the $AlN/WS_{2}$ bilayer. As for $WS_{2}$, the inclusion of the attractive e–h interaction on top of the GW calculations moves the absorption spectrum back towards lower energy and produces new peaks due to strongly bound excitons (orange solid lines in Figure 4). Hence, the $AlN/WS_{2}$ HT shows substantial adsorption from visible to deep UV light. Its spectrum is very similar to the $WS_{2}$ free standing one, but slightly blue shifted.

In order to understand the origin of the peaks and of the blue shift, we analyze the bands involved in the transitions.

Since, as mentioned, there is an interchange of the ordering of the GW bands, in the following, when we talk about the electronic bands involved in the excitonic transitions, we refer to the original DFT order. This is to avoid confusion in the analysis of the excitons. The first direct exciton at 2.15 eV is dark and double degenerate, and it is mainly due to the transition between the HOMO-1 and the LUMO at the **K** point, with a binding energy of 0.64 eV. The first visible exciton, also twice degenerate, appears at 2.21 eV and is characterized by a binding energy E${}_{b}$ = 0.59 eV. It involves the same bands as the two lowest dark excitons. The exciton binding energy is smaller than that of the isolated $WS_{2}$ (0.64 eV) due to the presence of the AlN layer that increases the screening of the electron–hole interaction. This reduction of the binding energy contributes to the blue shift of the excitons. The peak around 2.65 eV belongs to the transition at K between the third band below the highest valence band (HOMO-2) and the second conduction band (LUMO+1), and it also has double degeneration. An analysis of the first 10 excitons (most of them degenerate) of the $AlN/WS_{2}$ in terms of valence–conduction transitions shows that all correspond to an excitation involving only $WS_{2}$ states. In fact, the lowest inter-layer states between AlN and $WS_{2}$ appear at an energy corresponding to the $A^{\prime }$ states around 2.40 eV. These states are not visible in the spectra due to their very small dipole moment. By comparing the $AlN/WS_{2}$ spectra with those of the isolated AlN and $WS_{2}$ (see Figure 4), we can conclude that the dominant contribution comes from $WS_{2}$. The AlN contribution is limited to the high-energy region due to the large optical gap of about 4.6 eV [18] but indirectly affects the $WS_{2}$ through the increase of the screening and, as we discuss in the next section, through the lattice mismatch.

#### 3.3.1. Effect of the Substrate

As shown in Figure 4, lower panel, the first exciton in $AlN/WS_{2}$ is blue-shifted by about 0.1 eV with respect to the first exciton of isolated $WS_{2}$. This is a puzzling result since several experiments and theoretical calculations show an opposite trend [54,58]: an increasing of screening tends to red-shift the optical spectra. However, care must be taken when directly comparing two spectra. Indeed, several (interconnected) effects are responsible for the final change in the spectra (Figure 4): the variation of the screening, the variation of the direct gap and the change in the lattice constant. In order to test and disentangle these two effects, we also simulate an $AlN/WS_{2}$ bilayer forcing the lattice constant to be the same as the isolated $WS_{2}$ monolayer (a = 3.16 Å). In this case, we find that the direct gap is reduced by −0.05 eV with respect to the isolated $WS_{2}$, due to the additional screening generated by the presence of the AlN layer. On the other hand, as we have discussed before (see Table 1), the full relaxation of the bilayer induces a small compressive strain on $WS_{2}$ (a = 3.142 Å) in the HT. This increases the direct gap by 0.09 eV with respect to the unrelaxed bilayer (constrained to have a = 3.16 Å) and leaves the binding energy unchanged. This means that the variation of the lattice constant induces a change in the gap larger than the one due to AlN layer screening. The small strain applied has then essentially no effect on the excitonic binding energy, as is consistent with other works [59]. To summarize, this explains why the fully relaxed heterostructure has a larger GW gap than the isolated $WS_{2}$, even in the presence of a larger screening. This finding, combined with the reduction of the excitonic binding energy due to the substrate (*AlN*), explains the blue-shift of the first exciton in Figure 4.

#### 3.3.2. Exciton Model

Finally, it is worthwhile to notice that a simple 2D excitonic model based on the Rytova–Keldysh approach [60,61,62,63] would give reasonable results, predicting the trend of the excitonic binding energies close to the ab-initio values.

Within this model, the Hamiltonian that describes the interaction of an electron–hole pair in a homogeneous 2D sheet with parabolic bands is given by
(2)$$\left\{E_{g}-\frac{\hbar^{2}}{2\mu }\nabla_{\rho }^{2}+\hat{W}\left(\rho \right)\right\}\phi_{0}\left(\rho \right)=E_{0}\phi_{0}\left(\rho \right)$$
with μ exciton reduced mass and $\hat{W}\left(\rho \right)$ the statically screened electron–hole attraction potential:(3)$$W\left(\rho \right)=-\frac{\pi e^{2}}{2\rho_{0}}\left[H_{0}\left(\frac{\rho }{\rho_{0}}\right)-N_{0}\left(\frac{\rho }{\rho_{0}}\right)\right].$$
where $H_{0}$ is the Struve function, $N_{0}$ is the Neumann Bessel function of the second kind, and $\rho_{0}$ is the screening radius: $\rho_{0}=2\pi \alpha_{2D}$ with $\alpha_{2D}$ is the static electronic sheet polarizability.

Although the error is quite large (see Table 3), this method is confirmed to be a cheap, fast way to obtain qualitative results. The simplicity of this method relies on the fact that the effective masses and polarizabilities are the only needed ingredients, and both can be easily obtained within DFT.

## 4. Conclusions

In summary, we have presented a study on a novel $AlN/WS_{2}$ vdW heterostructure. Its negative formation energy as well as its small lattice mismatch between the constituent monolayers suggests that it should be possible to synthesize it. The analysis of the band alignment and decomposed partial charge density of the heterostructure shows that a calculation at the DFT level is not sufficient to identify the character of the heterostructure. The GW inverts the band order, transforming the heterostructure from type II to type I. The subsequent Bethe–Salpeter calculation of optical excitations confirms this result. All the lowest optical excitions, dark or bright, belong to the $WS_{2}$ monolayer part. These results suggest that a simple identification of heterostructures by means of local or semi-local functional in DFT may fail when compared with more advanced methods such as GW plus Bethe–Salpeter or hybrid functionals.

Finally, the optical spectrum of the heterostructure is very similar to that of the isolated $WS_{2}$, with small differences arising from the presence of the substrate, which acts through the strain and through the screening. We demonstrate that the AlN substrate modifies only slightly the electronic and optical properties of $WS_{2}$ and therefore could be employed as an excellent alternative insulating substrate, acting as a gate dielectric material, to the most commonly used h-BN [18].

## Figures and Tables

**Figure 1 materials-15-08318-f001:**
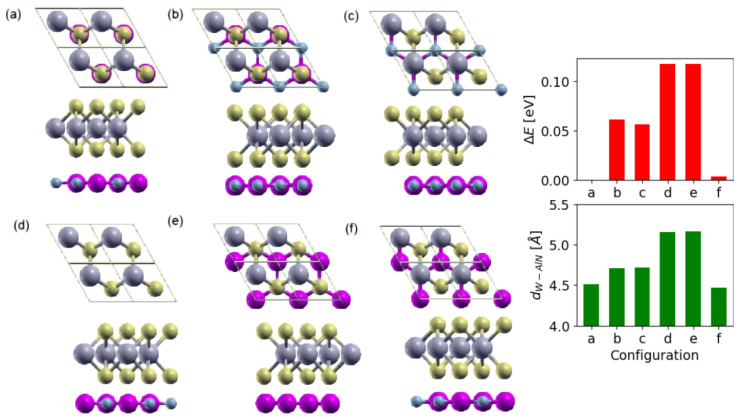
Top and side views of $AlN/WS_{2}$ heterostructures in different rotation angles: (**a**) 0${}^{\circ }$, (**b**) 60${}^{\circ }$, (**c**) 120${}^{\circ }$, (**d**) 180${}^{\circ }$, (**e**) 240${}^{\circ }$ and (**f**) 300${}^{\circ }$. Small light-blue balls indicate N atoms, Al atoms are in violet and gray and yellow indicate W and S atoms, respectively. On the right, we report the energy of each configuration with respect to the **(a)** one and the distance $d_{W-AlN}$.

**Figure 2 materials-15-08318-f002:**
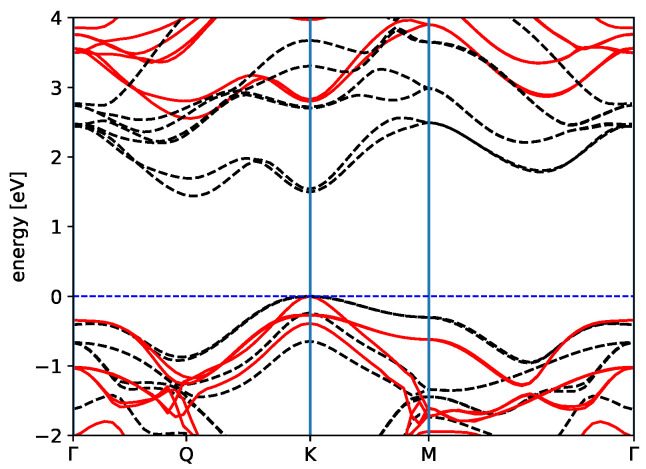
Band structure for the relaxed $AlN/WS_{2}$ vdW heterostructure along the high-symmetry path of the first Brillouin zone calculated within DFT (dashed black) and G${}_{0}$W${}_{0}$ (continuous red) level of approximation, taking into account the spin-orbit interaction. Notice that the minimum of the conduction bands appears between Γ and K, close to the Q point, and together with the top of the valence band at K forms the indirect band gap. The zero energy is set at the top of the valence bands in both the DFT and G${}_{0}$W${}_{0}$ calculations.

**Figure 3 materials-15-08318-f003:**
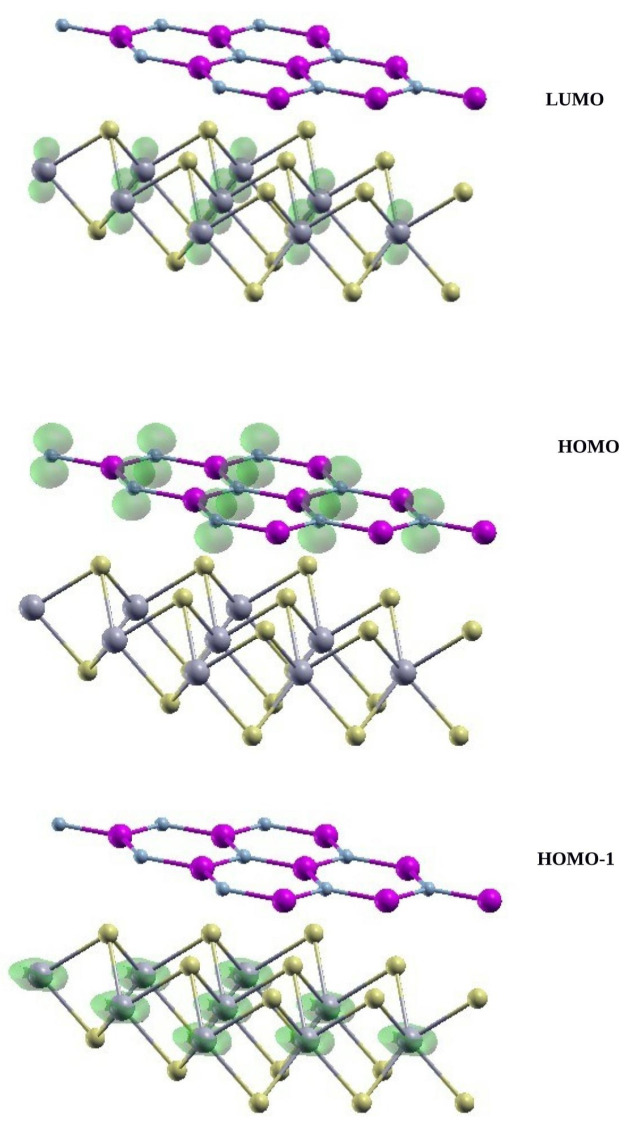
Band-decomposed charge density of the LUMO (**top**) and HOMO (**middle**) and HOMO-1 (**bottom**) at the DFT level for $AlN/WS_{2}$ vdW heterostructure at K point. The LUMO at Q, not reported in the figure, is very similar to the LUMO at K.

**Figure 4 materials-15-08318-f004:**
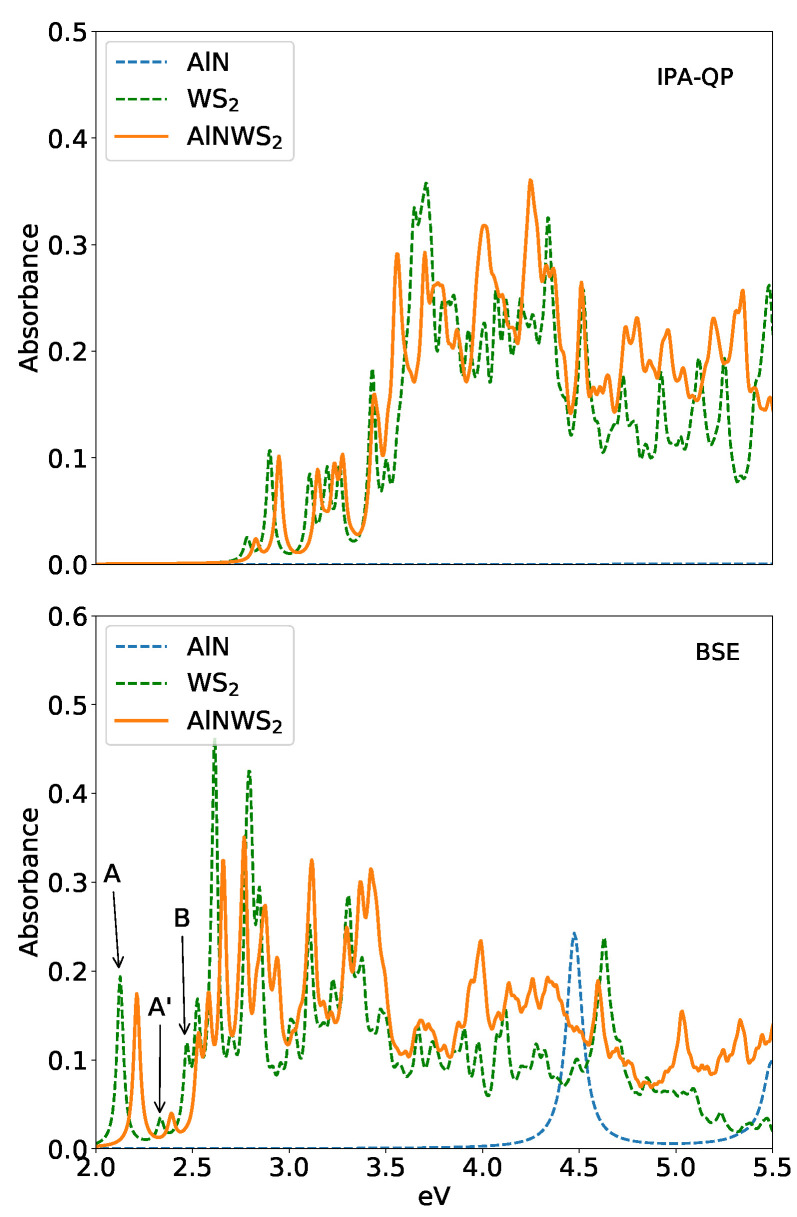
Absorbance of $AlN/WS_{2}$ heterostructure, and of isolated AlN and $WS_{2}$, calculated within the independent-quasi-particle approach G${}_{0}$W${}_{0}$ level (**top panel**) and Bethe–Salpeter equation (**bottom panel**).

**Figure 5 materials-15-08318-f005:**
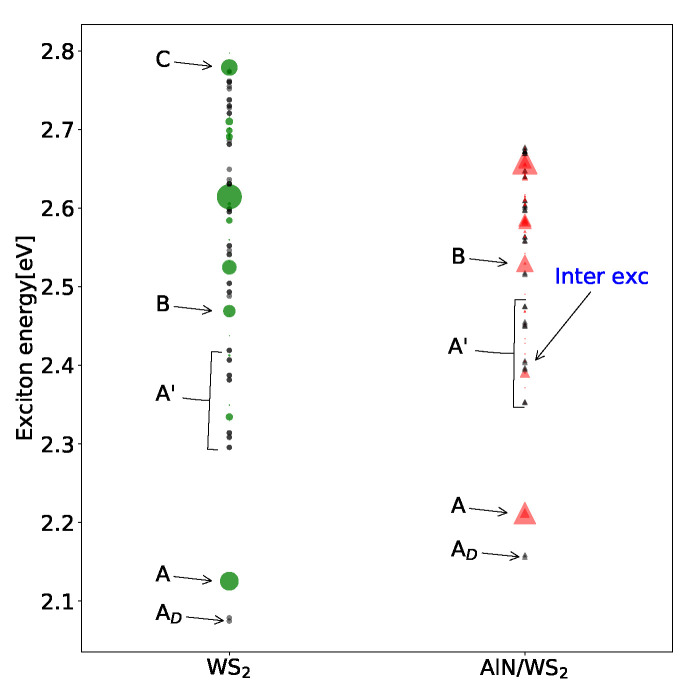
Schematic representation of the exciton level in the $WS_{2}$ layer and $AlN/WS_{2}$ bilayer. Here, we report the excitons at zero momentum—those responsible for the optical absorption. The size of the dots is proportional to the dipole of the different excitons. Dark states are represented by black dots. In the $AlN/WS_{2}$ case, we also report the position of the first inter-layer exciton. The nomenclature of excitonic states $A_{D}$ (dark spin forbidden), *A* (bright), *B* (bright), *C* (bright) is consistent with literature on TMDs.

**Table 1 materials-15-08318-t001:** Energy difference ΔE (eV), layer distance $d_{W-AlN}$, L${}_{W-S}$ and L${}_{Al-N}$ bond lengths of the different configurations of $AlN/WS_{2}$ heterostructures calculated by DFT-PBEsol including vdW correction. Notice that for the L${}_{W-S}$ distance, we consider the S atom towards the *AlN* layer.

conf.	ΔE (eV)	*a* (Å)	$d_{W-\mathrm{AlN}}$ (Å)	L${}_{W-S}$ (Å)	L${}_{\mathrm{Al}-N}$ (Å)
a	0	3.142	4.509	2.405	1.814
b	0.0613	3.140	4.705	2.405	1.813
c	0.0566	3.138	4.717	2.406	1.812
d	0.1176	3.137	5.158	2.404	1.811
e	0.1179	3.137	5.164	2.405	1.811
f	0.0032	3.144	4.469	2.403	1.815

**Table 2 materials-15-08318-t002:** Direct **K-K** and indirect **K-Q** band gaps for the three system at the G${}_{0}$W${}_{0}$ level. DFT results are given in parentheses. $E_{b}$ is the binding energy of the lowest bright exciton. The third row of the table refers to the relaxed bilayer, whereas the last row refers to the $AlN/WS_{2}$ HT forced to keep the same lattice parameter as the isolated $WS_{2}$.

System	Dir. Gap (eV)	Ind. Gap (eV)	E${}_{b}$
*AlN* (a = 3.11 Å)	6.27 (3.65)	5.61 (2.97)	1.79
$WS_{2}$ (a = 3.16 Å)	2.76 (1.64)	2.60 (1.64)	0.64
$AlN/WS_{2}$ (a = 3.142 Å)	2.80 (1.50)	2.56 (1.44)	0.59
$AlN/WS_{2}$ (a = 3.16 Å)	2.71 (1.38)	2.56 (1.38)	0.59

**Table 3 materials-15-08318-t003:** The excitonic binding energies of the first bright excitons calculated within the BSE. In parentheses, the values obtained within the 2D excitonic model are reported. In the last column, we report also the excitonic radius $r_{exc}$ obtained using the same model. In the second column are reported the hole and electron effective masses, m${}^{*}$, while in the third one are shown the values of the real part of the static polarizability.

	m${}_{h}^{*}$/m${}_{e}^{*}$ (m${}_{e}$)	Re $\alpha_{2D}$ (a.u.)	E${}_{b}$ (eV)	r${}_{\mathrm{exc}}$ (Å)
*AlN*	1.5/0.66	2.2	1.79 (2.3)	3.4
$WS_{2}$	0.42/0.44	13.0	0.64 (0.5)	11.4
$AlN/WS_{2}$	0.41/0.38	15.6	0.59 (0.4)	12.8

## Data Availability

Not applicable.

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
