# Peer review of "Interlayer and Intralayer Excitons in AlN/WS2 Heterostructure"

_materials, 2022, doi:10.3390/ma15238318_

Round 1

Reviewer 1 Report

    In the following work the Authors study electronic and optical properties of AlN/WS2 heterostructure using DFT+GW+BSE methodology. First, the Authors find energetically favorable stacking and then discuss the electronic structure at the DFT PBEsol level (with SOC). The main finding of this article is that G0W0 corrections “invert” predicted valence band ordering and additionally, the excitonic spectrum is consistent with this inversion, concluding that the AlN/WS2 heterostructure is of type 1 for both electronic and optical gaps. I find this discussion very interesting.

My main question is about predicted accuracy of this calculation – it seems that he Authors use parameters suggested by PRB 78, 085125 (2008), e.g. Fig. 1. However, this paper shows GW correction in the 3D system, and in 2D systems (e.g. for MoS2, PRB 93, 235435 (2016)) the corresponding convergence is probably order of magnitude slower in both dielectric cutoff and number of bands. How is using a 5 Ha cutoff and 500 bands justified?

My second comment is related to the inversion of bands (DFT before and after GW correction) – the Authors write only that “This band exchange is due to the different nature of the top valence bands belonging to the AlN and WS2 that acquire a different GW correction as it happens, often, in molecular systems [47].”. I think that this process deserves more physical explanation, e.g. is it related to different orbital nature of valence band states (know e.g. form ZnO, as in PRL 105, 146401 (2010))? What is the exact mechanism here?

My third comment relates to comparing AlN and h-BN as substrates for WS2 – Authors mention in several places that AlN has potential for studying stronger correlations due to smaller screening, but monolayer AlN is not compared with monolayer hBN (e.g. polarizabilities, A exciton binding energies etc.)? Perhaps at least simple comparison for effective 2D model with R-K screening would be useful?

Minor physical questions/comments:
1)    Line 75: Are k-mesh for BSE also 33x33x1 as for G0W0?
2)    Line 79: What are the parameters for relaxation studies?
3)    Figure 1: I suggest increasing resolution of this figure and adding schematically lattice parameters from Table 1 (a, d_W-AlN, L_W-S, L_Al-N), e.g. in Fig. 1(a).
4)    Line 119: It would be clearer if the Authors added that they discuss splitting of, presumably, conduction bands (the same comment for lines 138 and 141).
5)    Line 137: I suggest adding coordinates of the K point similar to the Q point.
6)    Figure 5: Are all excitonic states, shown in Fig. 5, calculated at the K point assuming direct-only (exciton total momentum = 0 assumption) transitions? Probably specifying this in the caption would help.
7)    Line 235: I suggest writing 2D exciton model with Rytova-Keldysh screening to set the notation for m*, alpha_2D and clarify how this effective model is solved to produce Eb. I would also add r_exc. from it for comparison in the last column of Table 3.
8)    Line 253: I think this line is a bit misleading, BSE does not confirm GW band inversion in heterostructure, it rather predicts that the studied heterostructure has ‘type I optical gaps’, fact that is not immediately trivial in the system studied.

Minor technical comments:
1)    Author affiliations: ETSF affiliation is not displaying correctly
2)    Line 48: ALD not defined
3)    Line 63 vs line 72: energy cutoff in 120Ry (which perhaps already sets the number of plane waves) vs 10 000 plane waves
4)    Line 116: HOMO first used here, not in line 143, LUMO not defined
5)    Line 236: Keldysh
6)    Line 241: Section probably copied from template and not intended by the Authors.
7)    Line 273: Several abbreviations not included (TMD, DFT, PV, LED, ML, LUMO, HOMO, MBPT, PDOS, SOC)

Author Response

Here our reply to the first referee

Reviewer 2 Report

The manuscript titled “Interlayer and Intralayer excitons in AlN/WS2 heterostructure” contains some valuable and interesting results on excitonic properties, which have been calculated using DFT calculations. The reported findings merit due consideration for its publication; however, the results can be organized and presented with a higher clarity for a better understanding. The article is publishable upon proper/moderate revision. The authors may carefully address the following concerns before their manuscript can be considered further for publication.

1.      There has been a large number of recent studies on electronic and various emergent properties of heterostructures in 2D materials, e.g., Applied Surface Science 563 (2021) 150304,  Physica E 130 (2021) 114674, ACS Appl. Mater. Interfaces 12 (2) (2020) 3114–3126. The authors can consider including some of these current research articles in the Introduction of the paper in the interest of a little more systematic literature review on heterostructures of 2D materials. Moreover, a methodical comparison with earlier reports, wherever possible, will help to bring out or highlight the novelty in this work.

2.      The authors may address the stability of the AlN/WS2 heterostructures, say, lattice dynamical stability from the absence of negative frequencies in the phonon dispersion /mechanical stability from the satisfaction of Born-Huang elastic stability criteria)/thermal stability based on molecular dynamics simulation.

3.      How does the stacking energy of the studied heterostructure vary with interlayer distance?

4.      What do the authors intend to convey through the section 4 (Discussion)? It’s not quite clear. The authors may improve upon this part.

5.      Which van der Waals (vdW) dispersion correction (D, D2 or D3) has been employed in the description of AlN/WS2 heterostructure?

Author Response

Here the replies to the second referee
